# Evolution of Highly Biocompatible and Thermally Stable YVO_4_:Er^3+^/Yb^3+^ Upconversion Mesoporous Hollow Nanospheriods as Drug Carriers for Therapeutic Applications

**DOI:** 10.3390/nano12152520

**Published:** 2022-07-22

**Authors:** Eluri Pavitra, Hoomin Lee, Seung Kyu Hwang, Jin Young Park, Young-Kyu Han, Ganji Seeta Rama Raju, Yun Suk Huh

**Affiliations:** 1Department of Biological Engineering, Biohybrid Systems Research Center (BSRC), Inha University, Incheon 22212, Korea; pavitra@inha.ac.kr (E.P.); hmlee8907@inha.ac.kr (H.L.); 22152213@inha.edu (S.K.H.); 2Department of Electrical, Electronics and Software Engineering, Pukyong National University, Yongdang Campus, Busan 48547, Korea; pjy0329@pknu.ac.kr; 3Department of Energy and Materials Engineering, Dongguk University-Seoul, Seoul 04620, Korea; ykenergy@dongguk.edu

**Keywords:** mesoporous hollow nanospheriods, UC luminescence, color tunability, fluorescence imaging, antitumor activity

## Abstract

In recent times, upconversion nanomaterials with mesoporous hollow structures have gained significant interest as a prospective nano-platform for cancer imaging and therapeutic applications. In this study, we report a highly biocompatible YVO_4_:1Er^3+^/10Yb^3+^ upconversion mesoporous hollow nanospheriods (YVO_4_:Er^3+^/Yb^3+^ UC-MHNSPs) by a facile and rapid self-sacrificing template method. The Rietveld analysis confirmed their pure phase of tetragonal zircon structure. Nitrogen adsorption–desorption isotherms revealed the mesoporous nature of these UC-MHNSPs and the surface area is found to be ~87.46 m^2^/g. Under near-infrared excitation (980 nm), YVO_4_:Er^3+^/Yb^3+^ UC-MHNSPs showed interesting color tunability from red to green emission. Initially (at 0.4 W), energy back transfer from Er^3+^ to Yb^3+^ ions leads to the strong red emission. Whereas at high pump powers (1 W), a fine green emission is observed due to the dominant three-photon excitation process and traditional energy transfer route from Er^3+^ to Yb^3+^ ions. The bright red light from the membrane of HeLa cells confirmed the effective cellular uptake of YVO_4_:Er^3+^/Yb^3+^ UC-MHNSPs. The resonant decrease in cell viability on increasing the concentration of curcumin conjugated YVO_4_:Er^3+^/Yb^3+^ UC-MHNSPs established their excellent antitumor activity. Therefore, the acquired results indicate that these YVO_4_:Er^3+^/Yb^3+^ UC-MHNSPs are promising drug carriers for bioimaging and various therapeutic applications.

## 1. Introduction

Currently, the aid of mesoporous hollow nanomaterials in multimodal imaging and drug delivery has gained substantial attention owing to its high drug loading efficiency, long circulation times, and sustained drug release behavior [1,2,3]. So far, several research groups have focused on and established the potential of surface functionalized mesoporous silicates for drug delivery applications because of their higher surface area and straight narrow channels, which allows the high adsorption of drugs into their texture [4,5,6,7]. However, there are still considerable disadvantages, including toxicity and weak permeability into the cellular nucleus [8]. It is also a noticeable fact that, as compared to near-infrared (NIR) light, ultraviolet (UV) light source damages human organs. In addition, the cell culture media and some of the drugs show autofluorescence under UV–vis excitations [9,10]. In addition, the upconversion (UC) nanomaterials, which converts NIR light (808, 915, and 980 nm) to UV and visible light, exhibit innovative pharmacokinetics without a background autofluorescence [11,12,13]. Therefore, in order to avoid the mentioned shortcomings and enable the safest and most efficient drug delivery carriers, the current research is focused on the integration of UC luminescence properties with mesoporous hollow nanoparticles.

Among different rare-earth-based host matrices, yttrium vanadate (YVO_4_) has received abundant interest as an ideal host material owing to its exceptional characteristics such as excellent chemical and thermal stability, outstanding optical enactment, and high luminescence quantum yield [14,15]. Since the first production of the YVO_4_ host matrix, extensive studies have been conducted to establish its optical properties using various dopant ions because the D_2d_ point symmetry of Y^3+^ ions provides an idle doping site for lanthanide ions [16]. For instance, YVO_4_:Eu^3+^ is established as a viable red phosphor for cathode ray tubes and an inorganic scintillator, whereas its single crystals are proven as an excellent laser host materials [17,18,19,20,21]. Likewise, there are few reports available on the UC luminescence properties of YVO_4_ material and its application as a temperature sensor [22]. Until now, many researchers focused on the synthesis procedure and photoluminescence studies of YVO_4_:Eu^3+^ nanophosphors. However, no reports were available on biocompatibility and antitumor activity of YVO_4_:Er^3+^/Yb^3+^ mesoporous nanoparticles so far.

In the present study, we developed a highly stable and extremely biocompatible YVO_4_:1Er^3+^/10Yb^3+^ UC mesoporous hollow nanospheriods (YVO_4_:Er^3+^/Yb^3+^ UC-MHNSPs) as potential drug carriers by a simple and rapid self-sacrificing template method. The formation mechanism and the crystalline nature of YVO_4_:Er^3+^/Yb^3+^ UC-MHNSPs were explained through morphological and structural studies. Depending on the input laser pump power (980 nm), these UC-MHNSPs exhibited red and green emissions. The reason behind this color tunability was discussed with the help of a number of excited photons and the energy transfer pathways among Er^3+^ and Yb^3+^ ions. HeLa cells were used to assess the cytotoxicity, permeability, and cellular uptake of these YVO_4_:Er^3+^/Yb^3+^ UC-MHNSPs, and curcumin was used as a modal drug to evaluate their antitumor activity.

## 2. Materials and Methods

### 2.1. Synthesis of Monodisperse Y(OH)CO_3_:1Er^3+^/10Yb^3+^ Nanospheres

Y_(1−x−y)_(OH)CO_3_:xEr^3+^/yYb^3+^ nanospheres (YC:Er^3+^/Yb^3+^ NSs) were fabricated by a modified urea-based homogeneous precipitation (UHP) method according to our earlier report [23]. All the reagents (analytical grade) were from Sigma-Aldrich (Gangnam-gu, Seoul, South Korea) and used without any additional purification. In a typical process, required amounts of yttrium nitrate [Y(NO_3_)_3_∙6H_2_O], erbium nitrate [Eu(NO_3_)_3_∙5H_2_O], ytterbium nitrate [Yb(NO_3_)_3_∙5H_2_O], and urea (10 mg/mL) were dissolved in 100 mL of de-ionized (DI) water. After 2 h of vigorous stirring, 20 µL of tetrapropylammonium hydroxide (TPAOH) was added and the magnetic stirring was continued for 30 min. Eventually, the colloidal solution was moved to silicone oil bath, and the reaction temperature was fixed to 90 °C for 10 h. After that, the YC:Er^3+^/Yb^3+^ NSs were gathered by using centrifugation (6000× *g* rpm/5 min) technique and washed multiple times with ethanol and DI water.

### 2.2. Fabrication of YVO_4_:1Er^3+^/10Yb^3+^ UC-MHNSPs

To fabricate the UC-MHNSPs, the obtained precursor (YC:Er^3+^/Yb^3+^ NSs) was dispersed into 40 mL of triple-distilled DI water and sonicated for 30 min to obtain the homogeneous solution. In another beaker, required amount of ammonium metavanadate [NH_4_VO_3_], ammonium fluoride [NH_4_F], and 120 µL of 2 M hydrochloric acid (HCl) were dissolved in 30 mL DI water. Then, the YC:Er^3+^/Yb^3+^ NSs were added dropwise to the vanadium solution under continuous stirring. After 30 min of stirring, the mixture was transformed into a Teflon liner and the autoclave temperature was fixed to 180 °C for 10 h. Once the autoclave temperature was cooled down to normal room temperature, the residue was gathered by centrifugal separator and washed numerous times with DI water and ethanol. After drying the sample in an oven (75 °C) overnight, the final form of the YVO_4_:Er^3+^/Yb^3+^ UC-MHNSPs was ready for further characterizations. All the characterization techniques were presented in Appendix A.

### 2.3. Cell Culture and Cytotoxicity Test

HeLa cell lines (ATCC, Manassas, VA, USA) were grown in 24 well plates (at a density of 5 × 10^4^ cells/well) and cultured for 24 h and then incubated with different quantities (50, 25, 12.5, 6.25, and 3.125 ug/mL) of YVO_4_:Er^3+^/Yb^3+^ UC-MHNSPs in a CO_2_ incubator at 37 °C for another 24 h. After that, HeLa cell lines were washed with Dulbecco’s phosphate-buffered saline (DPBS) 5 times, and then the cell viability was measured by cell count kit-8 (CCK-8) based on the manufacturer’s protocol.

### 2.4. Drug Loading and In Vitro Releasing Protocol

For drug loading, 9 mg curcumin was dissolved in ethanol (9 mg/900 µL), and 45 mg YVO_4_:Er^3+^/Yb^3+^ UC-MHNSPs were dissolved in 2 mL DI water. The curcumin and UC-MHNSPs were used in 1:5 ratio. After 20 min of sonication, the curcumin solution was added dropwise to YVO_4_:Er^3+^/Yb^3+^ UC-MHNSPs and allowed to mix using a rotator for 24 h. The curcumin conjugated sample was collected by centrifugation (5000× *g* rpm/5 min) after washed with DI water several times.

## 3. Results and Discussion

### 3.1. Morphological Evolution

Herein, the nontoxic and highly water soluble YVO_4_:Er^3+^/Yb^3+^ UC-MHNSPs were produced by a rapid self-sacrificing template method. Initially, the YC:Er^3+^/Yb^3+^ NSs of around 50 nm size were synthesized by a well-established UHP route where TPAOH was used as a size controlling agent. At high temperatures (about 90 °C), urea decomposes into OH^−^ and CO_3_^2−^ ions, and then combines with Y^3+^ ions to produce the uniform YC:Er^3+^/Yb^3+^ NSs and the respective scanning and transmission electron microscope (SEM and TEM) images are shown in Figure 1a–d. Here, the addition of TPAOH (20 µL) encourages the nucleation process rather than crystal growth and thus the NSs size was limited to 50 nm. After that, these YC:Er^3+^/Yb^3+^ NSs were allowed to react with NH_4_VO_3_ under hydrothermal conditions (180 °C for 1 h), where the Y^3+^ cations react with VO_4_^3−^ anions and produces the YVO_4_:Er^3+^/Yb^3+^ UC-MHNSPs as a result of the Kirkendall effect. Figure 1e–h displays the corresponding SEM and TEM images of YVO_4_:Er^3+^/Yb^3+^ UC-MHNSPs and the possible chemical reactions that happened during the synthesis were given here:
(1)
$$5Y{\left(NO_{3}\right)}_{3}\cdot 6H_{2}O+5CO{\left(NH_{2}\right)}_{2}\;\to \;5Y\left(OH\right)CO_{3}+2H_{2}O+21NO_{2}+7NH_{3}$$


(2)
$$NH_{4}VO_{3}+Y\left(OH\right)CO_{3}+H_{2}O\;\to \;YVO_{4}+NH_{4}^{+}+OH^{-}+CO_{2}$$


(3)
$$Y\left(OH\right)CO_{3}+3H^{+}\;\to \;Y^{3+}+CO_{2}+2H_{2}O$$


(4)
$$NH_{4}VO_{3}+H_{2}O\;\to \;NH_{4}^{+}+VO_{4}^{3-}+2H^{+}$$


(5)
$$Y^{3+}+VO_{4}^{3-}\;\to \;YVO_{4}$$


The high-magnification SEM and TEM images (Figure 1f,h) unveils the mesoporous hollow nature of the YVO_4_ nanoparticles with a uniform size of around 150 nm. The high-resolution TEM (HRTEM) image of YVO_4_:Er^3+^/Yb^3+^ UC-MHNSPs (inset of Figure 1k) displayed clear lattice fringes with d-spacing of 2.676 Å, which is well synchronized with the theoretical d-spacing value of 2.668 Å corresponding to (112) plane of YVO_4_ host matrix. The selected area electron diffraction (SAED) patterns presented in Figure 1j,l reveals that the YC:Er^3+^/Yb^3+^ NSs were amorphous in nature whereas the ring pattern overlaid with bright spots confirms the nanocrystalline feature of YVO_4_:Er^3+^/Yb^3+^ UC-MHNSPs, respectively. The d-spacings 2.518, 2.012, 2.676, 3.568, and 1.823 Å obtained from the SAED pattern were analogous to the (220), (103), (112), (200), and (312) planes of tetragonal structured YVO_4_ matrix, respectively.

Energy-dispersive X-ray spectroscopy (EDS) was conducted to study the chemical characterization of YC:Er^3+^/Yb^3+^ NSs and YVO_4_:Er^3+^/Yb^3+^ UC-MHNSPs, and the resulting EDS spectra are produced in Figure 1i,k. The X-ray peaks that appeared in the energy spectrum of YC:Er^3+^/Yb^3+^ NSs (Figure 1i) confirmed the presence of host element (Y) together with dopant elements (Er, and Yb), while the energy spectrum of YVO_4_:Er^3+^/Yb^3+^ UC-MHNSPs established the existence of V along with Y, Er, and Yb elements as shown in Figure 1k. The position of X-ray peaks in both the spectra suggests that most of the Y, Er, and Yb ions occupied the L-shell, and V and O elements engaged the K-shell of the energy spectrum, respectively. Figure 1m–q shows the elemental mapping of a single UC-MHNSP and the acquired images signifying that all the elements (host and dopant) were uniformly distributed within the particle, which will help to enhance the luminescence properties of the YVO_4_ matrix.

Further, the formation mechanism of UC-MHNSPs can be explained through the Kirkendall effect, which describes the dissimilar diffusion rates of two elements in a multi-component system that leads to the formation of hollow and/or porous nanostructures [24,25]. When the YC:Er^3+^/Yb^3+^ NSs were added to the NH_4_VO_3_ solution, initially (within 5 min) the Y^3+^ ions at the surface of YC:Er^3+^/Yb^3+^ NSs will react with VO_4_^3−^ anions and forms a layer of YVO_4_ rod-like particles at the outer surface of the NSs and the corresponding chemical reactions are presented in Equations (3)–(5). As the reaction proceeds, the inner dissolved Y^3+^ cations tend to diffuse outwards, and simultaneously the VO_4_^3−^ anions in the solution try to diffuse inwards through NSs. When the outward diffusion rate of Y^3+^ cations was much quicker compared to the inward diffusion rate of VO_4_^3−^ anions, the voids will form within the NSs and the reaction will occur on or near the surface (or interface) which leads to the outward growth of the YVO_4_ hollow shell. Thus, the YVO_4_:Er^3+^/Yb^3+^ UC-MHNSPs were formed as a result of the Kirkendall effect and the schematic representation of the formation mechanism is displayed in Figure 2. Interestingly, all these reactions occurred within 1 h of time duration, which is very rapid, and this morphology continued even after 24 h. The corresponding low and high magnification SEM and TEM images of YVO_4_:Er^3+^/Yb^3+^ UC-MHNSPs at different reaction timings (4, 6, 12, and 24 h) were presented in the Appendix A, respectively.

### 3.2. Structural Analysis

The X-ray diffraction (XRD) technique was used to determine the crystalline nature of YC:Er^3+^/Yb^3+^ NSs and YVO_4_:Er^3+^/Yb^3+^ UC-MHNSPs, and the resulting diffraction patterns are presented in Figure 3a. The XRD profile of the precursor sample exhibited two broad bands with maxima at around 29° and 46°, indicating the amorphous nature of the YC:Er^3+^/Yb^3+^ NSs. Whereas a well indexed tetragonal phase was attained for YVO_4_:Er^3+^/Yb^3+^ UC-MHNSPs, which exactly matches with the previously reported reference pattern on the YVO_4_ host matrix with JCPDS card no. 76-1649 and the space group of I4_1_/amd_z_. Because of the similar ionic radii, no other impurity peaks were detected after the doping of Er^3+^ (0.89) and Yb^3+^ (0.868) ions in the Y (0.9) site, indicating that the crystalline structure of the YVO_4_ host lattice did not effected by the current doping level. Likewise, the effect of flux (NH_4_F) on the crystal structure of YVO_4_:Er^3+^/Yb^3+^ UC-MHNSPs was also verified. At 4 mmol concentration of NH_4_F, a pure tetragonal phase without any impurity peaks was observed whereas a mixed phase was observed for 8 mmol concentration of flux and the corresponding diffraction patterns are illustrated in Appendix A. In general, the excess of flux leads to the formation of mixed phases and thus 4 mmol NH_4_F was used as flux during the synthesis of YVO_4_:Er^3+^/Yb^3+^ UC-MHNSPs [26]. Figure 3b displays the diffraction patterns of YVO_4_:Er^3+^/Yb^3+^ UC-MHNSPs synthesized at various reaction timings from 5 min to 24 h. Interestingly, the diffraction pattern exactly matched the reference pattern (JCPDS No. 76-1649) and the witnessed increase in diffraction pattern intensity signified the increasing crystallinity of the particles. The well-known Scherrer equation was employed to evaluate the mean crystallite size (D):D_hkl_ = κλ/β cos θ
where k and λ represents the Scherrer constant and X-ray beam wavelength, while θ and β indicates the diffraction angle and full width at half maximum of the diffraction peak, respectively. The average crystallite sizes were calculated to be 23.6, 24.9, 25.7, 26.9, 32.9, 34.5, 35.5, and 37.4 nm for 5 min, 15 min, 30 min, 1 h, 4 h, 6 h, 12 h, and 24 h of reaction time, respectively. After the addition of flux (4 mmol NH_4_F), the mean crystallite size increased from 26.8 to 41.7 nm for YVO_4_:Er^3+^/Yb^3+^ UC-MHNSPs at 1 h of reaction time, signifying the increased crystallinity of the material. General Structure Analysis System (GSAS) software was used to execute the Rietveld refinement of the XRD pattern to validate the tetragonal zircon-type structure of YVO_4_:Er^3+^/Yb^3+^ UC-MHNSPs. Figure 3c presents the Rietveld fit of the YVO_4_:Er^3+^/Yb^3+^ UC-MHNSPs where the difference profile appeared to be nearly flat signifying a good quality fit between calculated and experimental profiles with decent agreement values (R_p_ = 2.32, R_wp_ = 2.96, and χ2 = 2.682). The obtained crystallographic data including atomic positions and occupancies are presented in Appendix A. The obtained unit cell parameters were: a and b = 7.1095 Å, c = 6.2844 Å, and volume (V) = 317.65 Å^3^ for YVO_4_:Er^3+^/Yb^3+^ UC-MHNSPs. Diamond 3.2, a crystal structure visualization software, was used to sketch the crystal structure of YVO_4_:Er^3+^/Yb^3+^ UC-MHNSPs and the resultant ideal unit cell model is displayed in Figure 3d. Generally, the YVO_4_ host lattice belongs to rare-earth orthovanadates (MVO_4_) with tetragonal zircon structure with space group I4_1_/amd_z_ (where Z = 4) [27]. This zircon-type structure is assembled from the chains of alternative edge-sharing [VO_4_]^3−^ tetrahedra and [YO_8_] bisdisphenoids which are growing parallel to the c-axis and laterally linked through edge-sharing [YO_8_] bisdisphenoids to build “zigzag” chains parallel to a-axis as shown in Figure 3d.

Brunauer–Emmett–Teller (BET) approach was adopted to quantify the surface area of YC:Er^3+^/Yb^3+^ NSs and YVO4:Er^3+^/Yb^3+^ UC-MHNSPs and the resultant N_2_ adsorption–desorption isotherms are shown in Figure 4. The acquired adsorption–desorption profiles of both the samples resembles the type IV isotherm accompanied by H_3_ type hysteresis loop, based on the IUPAC classifications [28,29]. The hysteresis loop appeared at high pressure region (ρ/ρ_0_ = 0.8 to 0.99) shown in Figure 4a indicates the presence of large pores (mesopores to macropores) in the YC:Er^3+^/Yb^3+^ NSs, whereas the wider hysteresis loop at partial relative pressure region (ρ/ρ_0_ = 0.4 to 0.8) illustrated in Figure 4b signifies the existence of mesoporous nature of the YVO4:Er^3+^/Yb^3+^ UC-MHNSPs with narrow sized pore distribution. The specific surface area of YC:Er^3+^/Yb^3+^ NSs and YVO4:Er^3+^/Yb^3+^ UC-MHNSPs were found to be 25.74 and 87.46 m^2^/g, respectively. Compared to the NSs, the surface area is 3.4-fold increased for MHNSPs. According to the Barrett–Joyner–Halenda (BJH) plot shown in Figure 4c, the pore size distribution was depicted prominently at around 3.8 nm along with average pore volume of 0.16 cm^3^/g for YVO_4_:Er^3+^/Yb^3+^ UC-MHNSPs, and the broad peak obtained at around 82 nm region belongs to the hollow structure of the particle. Thus, the BET analysis infers the mesoporous nature of our material.

Zeta potential was measured to study the dispersion stability of the YC:Er^3+^/Yb^3+^ NSs and YVO_4_:Er^3+^/Yb^3+^ UC-MHNSPs in DI water and the acquired results are shown in Figure 4d. As expected, both the NSs and UC-MHNSPs exhibited positive surface charge of 15.29 and 31.99 mV, respectively. Because YC:Er^3+^/Yb^3+^ NSs were synthesized by the well-known UHP method (where urea (CO(NH)_2_) produces NH_4_^+^ ions) and for the synthesis of YVO_4_:Er^3+^/Yb^3+^ UC-MHNSPs, NH_4_VO_3_ was taken as vanadium source and NH_4_F was used as flux. Thus, the excess of NH_4_^+^ ions present in the solvent causes the positive charge density on the surface of the UC-MHNSPs. Typically, the nanoparticles with Zeta potential lower than −30 mV or higher than +30 mV are considered as highly dispersible and are suitable for biomedical applications. Hence, our synthesized YVO_4_:Er^3+^/Yb^3+^ UC-MHNSPs with 31.99 mV Zeta potential can be used as an efficient drug carrier for biomedical applications.

### 3.3. Upconversion Luminescence Studies

It has been well established that, the Yb^3+^ ions absorb most of the incident NIR light and then resonantly donates their energy to the adjacent Er^3+^ ions. Because of the spectral overlap between the ^4^I_15/2_ → ^4^I_11/2_ transition of Er^3+^ ions and ^2^F_7/2_ → ^2^F_5/2_ transition of Yb^3+^ ions, the co-doping of Er^3+^/Yb^3+^ ions forms the most attractive activator/sensitizer pair to achieve the UC luminescence [30]. Figure 5a illustrates the luminescence spectra of YVO_4_:Er^3+^/Yb^3+^ UC-MHNSPs as a function of excitation pump power from 0.4 to 1 W under 980 nm laser excitation. All the emission spectra displayed a weak green emission band between 516 to 560 nm region with maxima at 522 and 546 nm along with an intense red emission band around 642 to 690 nm region with band maxima at 670 nm, which are analogous to the transitions from the emitting excited levels ^2^H_11/2_, ^4^S_3/2_, and ^4^F_9/2_ to the ground level ^4^I_15/2_ of Er^3+^ ions. It is worthwhile to mention that, at low pump powers of 0.4 W, the UC emission spectrum displayed only the red emission band and the green emission band intensity increased along with the red emission band by rising the input laser pump power from 0.5 to 1 W (Figure 5a). The comparison between emission spectra of YVO_4_:Er^3+^/Yb^3+^ UC-MHNSPs at low (0.4 W) and high (1 W) pump powers are presented in Figure 5b. The digital photos of red and green emissions at low and high pump powers are displayed as insets of Figure 5b. According to the literature, energy back transfer (EBT) from Er^3+^ ions to Yb^3+^ ions and cross-relaxation (CR) among adjacent Er^3+^ ions might be the reasons for the occurrence of a strong red emission band as compared to the green emission band [31,32].

The dependency of the integrated intensity ratio of red and green (R/G) emission bands versus laser pump power is presented in Figure 5c. The R/G values were decreased continuously (from 16.65 to 2.66) by rising the excitation power from 0.4 to 1 W. In general, the intensities of red and green emission bands depend on the amount of population in the excited states of Er^3+^ ions (^2^H_11/2_, ^4^S_3/2,_ and ^4^F_9/2_). In the present case of YVO_4_:Er^3+^/Yb^3+^ UC-MHNSPs, the intensities of both red and green emission bands constantly increased up to 1 W laser pump power without quenching. Compared to the red emission intensity, a strong increase was observed for green emission intensity due to the traditional energy transfer (ET) and non-radiative (NR) transitions, and hence the R/G ratio was decreased at high excitation pump powers. To understand the UC excitation process, the study of UC emission intensity (I) versus excitation pump power (P) is important and their relation is denoted by the following Equation: [33]

$$I\propto P^{n}$$

where *n* represents the number of pumping photons needed to populate the desired emitting states. Figure 5d shows the plots of log (*I*) vs log (*P^n^*) for green (^2^H_11/2 →_ ^4^I_15/2_) and red (^4^F_9/2 →_ ^4^I_15/2_) emission peaks, which formed a straight line. The slope values were calculated to be 1.32 and 3.09 for red and green emissions, suggesting the involvement of two-photon and three-photon excitation processes for red and green emissions, respectively.

The population process of emitting excited states and the proposed energy transfer routes in YVO_4_:Er^3+^/Yb^3+^ UC-MHNSPs are schematically visualized in Figure 6. Typically, ground state absorption (GSA), ET, and excited state absorption (ESA) are the three basic possible ways that are responsible to populate the excited levels in the UC mechanism [34]. Under 980 nm laser excitation, the Yb^3+^ and Er^3+^ ions in ^2^F_7/2_ and ^4^I_15/2_ states absorb the incident photons and are excited to ^2^F_5/2_ and ^4^I_11/2_ states as a result of GSA. As compared to Er^3+^ ions, Yb^3+^ ions absorb more NIR radiation and transfer their energy to ^4^I_11/2_ and ^4^F_7/2_ levels of Er^3+^ ions through the ET process. At the same time, the Er^3+^ ions in the ^4^I_11/2_ level are further excited to the ^4^F_7/2_ level by grabbing one more photon of 980 nm and this method is acknowledged as ESA. Hence, the ^4^F_7/2_ level is populated by GSA, ET, and ESA routes. Then, the excited Er^3+^ ions in the ^4^F_7/2_ level undergo NR transition to populate ^2^H_11/2_, ^4^S_3/2_, and then ^4^F_9/2_ levels. As a result of radiative transitions ^2^H_11/2_, ^4^S_3/2_ → ^4^I_15/2_, the relatively low intense green emission bands around 522 and 546 nm are achieved. However, the proposed three-photon excitation process for green emission occurred at high pump powers due to CR between adjacent Er^3+^ ions. Some of the Er^3+^ ions in the emitting excited levels (^2^H_11/2_, ^4^S_3/2_) can further be excited to the ^2^G_7/2_ level through the ET process and then relax to the ^4^G_11/2_ level by NR transition. Further, the emitting excited levels ^2^H_11/2_ and ^4^S_3/2_ were populated through the CR processes (^4^G_11/2_ → ^2^H_11/2_, ^4^S_3/2,_ and ^4^I_15/2_ → ^4^I_13/2_)) between Er^3+^ ions [35,36,37].

In the case of strong red emission, other than NR transition from ^2^H_11/2_, ^4^S_3/2_ levels to ^4^F_9/2_ level, there are two more possible ways to populate the^4^F_9/2_ level. Among them, the first one is ESA (^4^I_13/2_ + a photon (980 nm) → ^4^F_9/2_) or ET (^2^F_5/2_ + ^4^I_13/2_ → ^2^F_7/2_ + ^4^F_9/2_) process, where the level ^4^I_13/2_ is populated by NR transition from ^4^I_11/2_ level [38]. An another way to populate ^4^F_9/2_ level is CR process (^4^F_7/2_ → ^4^F_9/2_ + ^4^I_11/2_ → ^4^F_9/2_) among Er^3+^ ions and EBT process from Er^3+^ to Yb^3+^ ions (^4^S_3/2_ + ^2^F_7/2_ → ^4^I_13/2_ + ^2^F_5/2_) as shown in Figure 6 [39]. Therefore, the green emitting excited levels (^2^H_11/2_ and ^4^S_3/2_) are more populated when compared to the red emitting level (^4^F_9/2_), and thus an intense red emission band achieved for YVO_4_:1Er^3+^/10Yb^3+^ UC-MHNSPs. In addition, during the synthesis of YVO_4_:Er^3+^/Yb^3+^ UC-MHNSPs, (4 mmol) NH_4_F was used as a flux to enrich the emission properties. It is well established that a small amount of suitable flux enhances the crystallization of the material which eventually leads to the enhancement of its luminescence properties [26,40].

Figure 7a displays the temperature-dependent emission spectra of YVO_4_:Er^3+^/Yb^3+^ UC-MHNSPs under 980 nm laser excitation. On rising the temperature from 25 to 160 °C, the profile and band positions of green and red emissions did not alter, but the intensities were slightly decreased due to increased lattice vibrations. The inset of Figure 7a shows the luminescence intensity ratio (LIR) versus the temperature graph for thermally coupled red emission levels (I_657_/I_670_). Usually, the LIR is calculated for thermally coupled levels (^2^H_11/2_ and ^4^S_3/2_) of green emission to estimate the sensing behavior of the material. However, in the present case of YVO_4_:Er^3+^/Yb^3+^ UC-MHNSPs, red emission was more prominent than the green emission and thus we calculated LIR for stark levels of red emission band, which is almost flat [41,42]. Surprisingly, at an elevated temperature of 160 °C, around 87.5% emission intensity was acquired as compared to the room-temperature (25 °C) emission intensity, signifying the high thermal stability of these YVO_4_:Er^3+^/Yb^3+^ UC-MHNSPs. Likewise, the photostability was evaluated for these UC-MHNSPs under continuous laser irradiation for 20 min, and the resultant emission spectra under high pump power of 1 W are displayed in Figure 7b. Interestingly, 90% of the emission intensity was attained even after 20 min of continuous laser irradiation at the high input pump power of 1 W. Therefore, the obtained results suggest that our synthesized YVO_4_:Er^3+^/Yb^3+^ UC-MHNSPs are highly stable at elevated temperatures and longer irradiation times.

In addition, the LIR (I_657_/I_670_) increased gradually on rising the pump power from 0.4 to 1 W (Figure 7c), indicating the presence of laser-induced heat effect on the emission spectra of YVO_4_:Er^3+^/Yb^3+^ UC-MHNSPs. Generally, laser irradiation causes an increase in the local temperature of the sample as a result the emission intensity increases. Therefore, the laser-induced heat effect was estimated for YVO_4_:Er^3+^/Yb^3+^ UC-MHNSPs at low and high pump powers of 0.4 and 1 W, and the corresponding LIR (I_657_/I_670_) versus irradiation time plots are presented in Figure 7d. As shown in the figure, the change in LIR (I_657_/I_670_) was negligible at the low pump power of 0.4 W, as compared to the change in LIR (I_657_/I_670_) at the high pump power of 1 W, suggesting that the laser-induced heat effect was insignificant at low pump powers. Therefore, the input laser pump power was set to 0.4 W for biomedical applications to avoid the laser-induced heat effect.

### 3.4. In Vitro Imaging, Cell Viability, and Antitumor Activity

In vitro imaging studies were performed to study the permeability and translocation of YVO_4_:Er^3+^/Yb^3+^ UC-MHNSPs using human cervical carcinoma (HeLa) cell lines. Figure 8a–d shows the bright field and dark field images of the HeLa cells under 980 nm excitation with a fixed pump power of 0.4 W. Figure 8a,b shows the confocal microscope images of the nuclear stain (Hoechst 33342) and the red emission from YVO_4_:Er^3+^/Yb^3+^ UC-MHNSPs, respectively. The merged images displayed in Figure 8c,d confirmed the successful uptake of YVO_4_:Er^3+^/Yb^3+^ UC-MHNSPs by the cell lines. The bright red luminescence spots observed in the cytoplasm and nucleoplasm of the HeLa cell lines revealed the penetration capability of these UC-MHNSPs. The absence of background autofluorescence and the effective cellular uptake signify that these YVO_4_:Er^3+^/Yb^3+^ UC-MHNSPs can be used as promising live cell imaging agents for cancer imaging and therapeutic applications.

Likewise, the cytotoxicity of UC nanoparticles is one of the most important concerns for their use in biomedical applications. The HeLa cells’ feasibility was examined using Kit-8 cell count kit (CCK-8) assay after incubation with YVO_4_:Er^3+^/Yb^3+^ UC-MHNSPs for 24 h. Figure 8e shows the viabilities of HeLa cell lines treated with various concentrations (100, 50, 25, 12.5, and 6.25 µg/mL) of YVO_4_:Er^3+^/Yb^3+^ UC-MHNSPs. After 24 h of incubation, our synthesized UC-HMNSPs showed negligible effect on the activation and proliferation of HeLa cells even at higher concentrations (100 µg/mL), suggesting the non-toxic nature of these YVO_4_:Er^3+^/Yb^3+^ UC-MHNSPs. We also calculated the drug loading efficiency of YVO_4_:Er^3+^/Yb^3+^ UC-MHNSPs and the amount of curcumin loaded on YVO_4_:Er^3+^/Yb^3+^ UC-MHNSPs was estimated to be about 12%. Most of the time, the direct loading of drugs does not yield high loading efficiencies and surface functionalization is needed to improve their drug loading efficiency. In the present study, the curcumin was loaded on YVO_4_:Er^3+^/Yb^3+^ UC-MHNSPs particles without any surface functionalization and thus the acquired amount of curcumin loading was somewhat less.

Furthermore, the antitumor activity of curcumin conjugated YVO_4_:Er^3+^/Yb^3+^ UC-MHNSPs was studied on HeLa cells. The YVO_4_:Er^3+^/Yb^3+^@Curcumin composite showed a substantial effect on HeLa cells viability as shown in Figure 8f. Upon rising the concentration of YVO_4_:Er^3+^/Yb^3+^@Curcumin composite from 3.125 to 50 µg/mL, the HeLa cells’ viability decreased from 98.3 to 46.7%, indicating the drug releasing capability of YVO_4_:Er^3+^/Yb^3+^ UC-MHNSPs on HeLa cell lines. Generally, the drug carrier needs to protect the drug from degradation by enzymes and bio-environment before reaching the targeted tumor site and these YVO_4_:Er^3+^/Yb^3+^@Curcumin UC-MHNSPs exhibited an abundant antitumor effect on HeLa cell lines. Therefore, the obtained results indicate that these YVO_4_:Er^3+^/Yb^3+^ UC-MHNSPs have great potential as drug carriers due to their excellent dispersion stability, biocompatibility, and UC luminescence properties.

## 4. Conclusions

Highly stable, aqueous dispersible, and biocompatible YVO_4_:Er^3+^/Yb^3+^ UC-MHNSPs were successfully synthesized by a facile and rapid self-sacrificing templet method. The pure phase of the tetragonal zircon-type structure was confirmed by performing the Rietveld refinement analysis on the XRD patterns of YVO_4_:Er^3+^/Yb^3+^ UC-MHNSPs. The transformation of YC:Er^3+^/Yb^3+^ NSs into YVO_4_:Er^3+^/Yb^3+^ UC-MHNSPs was explained with the help of FE-SEM and FE-TEM images. The N_2_ adsorption–desorption isotherms confirmed the mesoporous nature of these materials and the specific surface area was found to be 87.46 m^2^/g. Interestingly, these YVO_4_:Er^3+^/Yb^3+^ UC-MHNSPs exhibited red emission at low pump powers and a fine green emission at high pump powers under NIR excitation. At low pump powers, EBT from Er^3+^ to Yb^3+^ ions and CR among Er^3+^ ions result in strong red emission. While at high pump powers, along with ET from Er^3+^ to Yb^3+^ ions, the three-photon excitation process takes place to populate the green emitting excited states (^2^H_11/2_, ^4^S_3/2_). These YVO_4_:Er^3+^/Yb^3+^ UC-MHNSPs exhibited around 87.5 and 90% emission intensity at an elevated temperature of 160 °C and continuous laser irradiation of 20 min, demonstrating the high thermal stability and photostability of these UC-MHNSPs. The bright red spots from the membrane of HeLa cells confirmed the effective cellular uptake of these YVO_4_:Er^3+^/Yb^3+^ UC-MHNSPs. After conjugating with curcumin, the HeLa cells’ viability was significantly decreased, demonstrating the excellent antitumor activity of these UC-MHNSPs. Based on these results, it is quite clear that these YVO_4_:Er^3+^/Yb^3+^ UC-MHNSPs have the potential to be used as bioimaging agents and drug delivery systems in various therapeutic applications.

## Figures and Tables

**Figure 1 nanomaterials-12-02520-f001:**
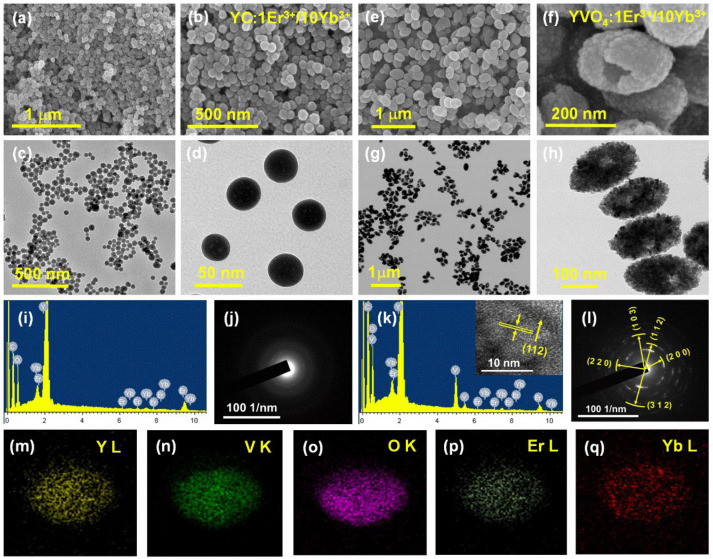
FE-SEM and FE-TEM images of (**a**–**d**) YC:Er^3+^/Yb^3+^ NSs and (**e**–**h**) YVO_4_:Er^3+^/Yb^3+^ UC-MHNSPs. (**i**,**k**) EDS spectrum and (**j**,**l**) SAED patterns of NSs and UC-HMNSPs, respectively, and the inset of (**k**) shows the corresponding HR-TEM image. (**m**–**q**) Elemental mapping of an individual YVO_4_:Er^3+^/Yb^3+^ UC-MHNSPs based on EDS spectrum.

**Figure 2 nanomaterials-12-02520-f002:**
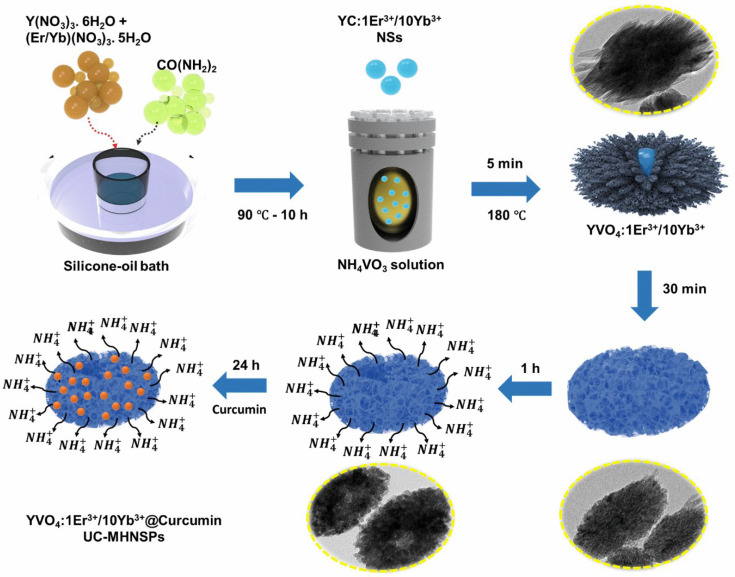
Schematic of the synthetic route and the transformation of YC:Er^3+^/Yb^3+^ NSs into YVO_4_:Er^3+^/Yb^3+^ UC-MHNSPs.

**Figure 3 nanomaterials-12-02520-f003:**
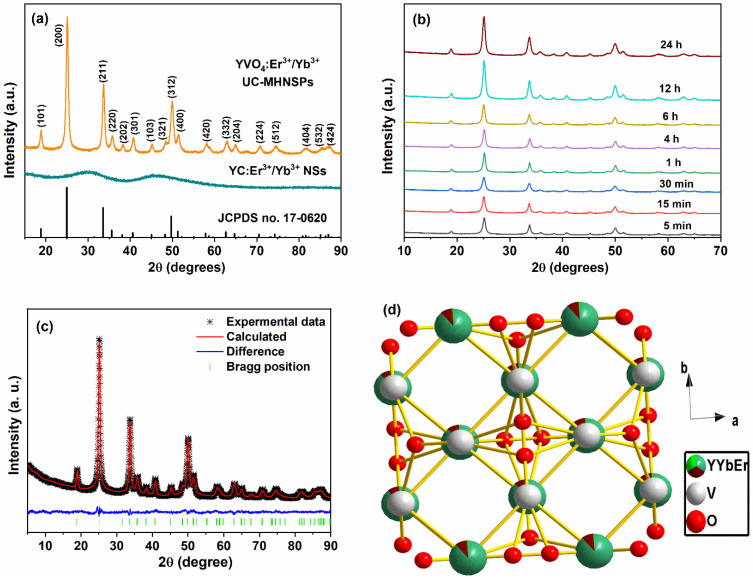
(**a**) XRD patterns of YC:Er^3+^/Yb^3+^ NSs and YVO_4_:Er^3+^/Yb^3+^ UC-MHNSPs, (**b**) diffraction patterns of YVO_4_:Er^3+^/Yb^3+^ UC-MHNSPs at various reaction timings, (**c**) Rietveld fit and (**d**) ideal unit cell of YVO_4_:Er^3+^/Yb^3+^ UC-MHNSPs.

**Figure 4 nanomaterials-12-02520-f004:**
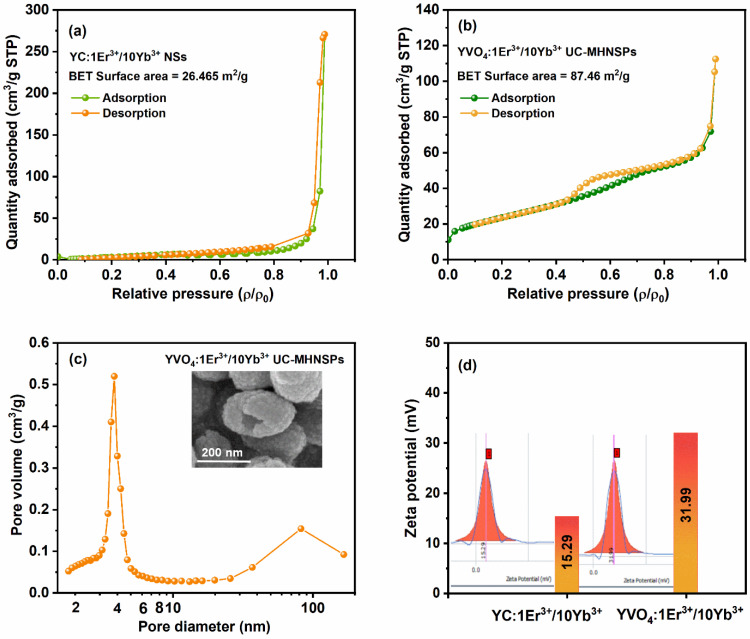
(**a**,**b**) Nitrogen adsorption desorption isotherms of YC:Er^3+^/Yb^3+^ NSs and YVO_4_:Er^3+^/Yb^3+^ UC-MHNSPs, (**c**) pore size distribution plot of YVO_4_:Er^3+^/Yb^3+^ UC-MHNSPs, and (**d**) particle zeta potential value and the related distribution graphs of NSs and UC-MHNSPs.

**Figure 5 nanomaterials-12-02520-f005:**
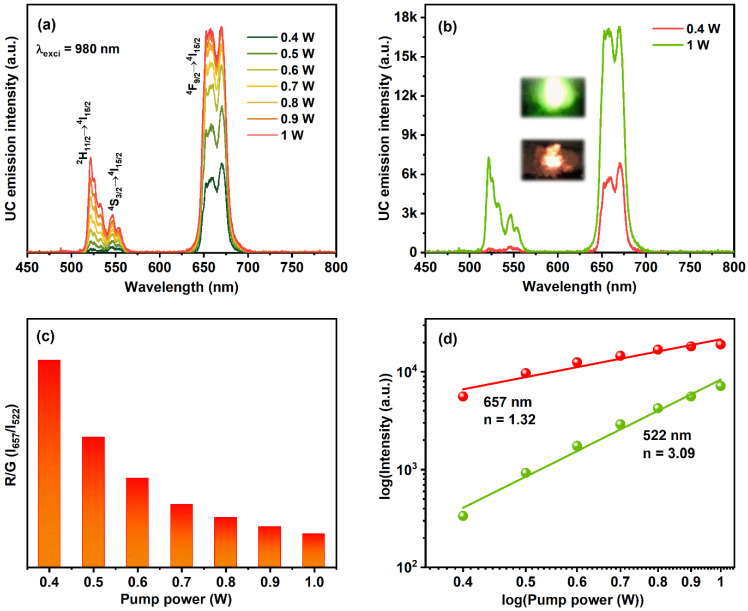
(**a**) UC luminescence spectra of YVO_4_:Er^3+^/Yb^3+^ UC-MHNSPs as a function of incident pump power under 980 nm excitation, (**b**) comparison between the emission spectra of YVO_4_:Er^3+^/Yb^3+^ UC-MHNSPs at low and high excitation powers and the insets show the corresponding digital photos. (**c**) R/G versus input pump power and (**d**) logarithmic plots of input laser pump power versus green and red emission intensities.

**Figure 6 nanomaterials-12-02520-f006:**
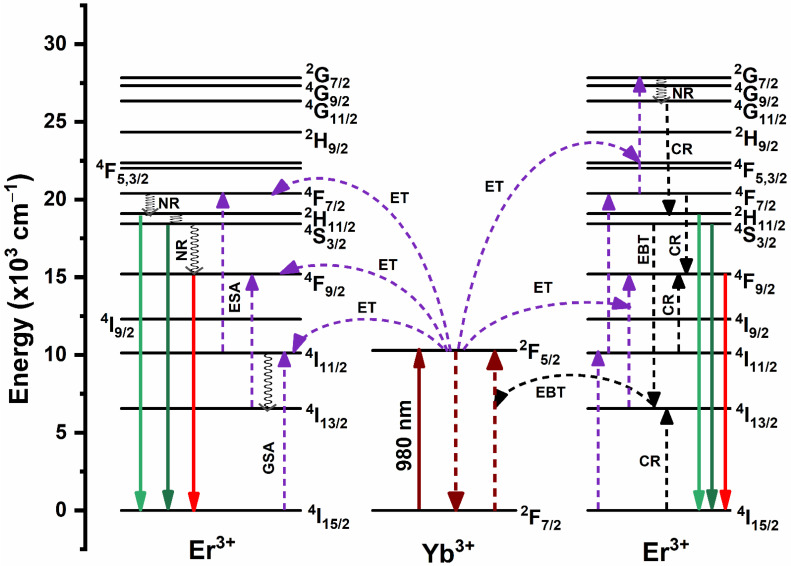
Schematic energy level illustration of Er^3+^ and Yb^3+^ ions, and the suggested UC excitation mechanism in YVO_4_:Er^3+^/Yb^3+^ UC-MHNSPs.

**Figure 7 nanomaterials-12-02520-f007:**
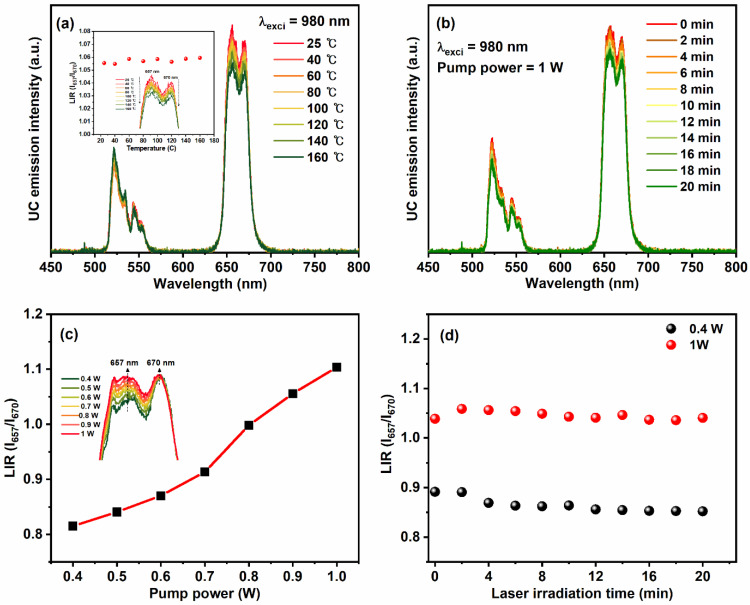
(**a**) Temperature-dependent emission spectra of YVO_4_:Er^3+^/Yb^3+^ UC-MHNSPs under 980 nm excitation and the inset displays the corresponding LIR (I_657_/I_670_) versus temperature plot. (**b**) UC emission spectra of YVO_4_:Er^3+^/Yb^3+^ UC-MHNSPs as a series of laser irradiation time, (**c**) a plot of LIR (I_657_/I_670_) versus incident laser pump power, and (**d**) LIR (I_657_/I_670_) versus laser irradiation time plots at low and high pump powers.

**Figure 8 nanomaterials-12-02520-f008:**
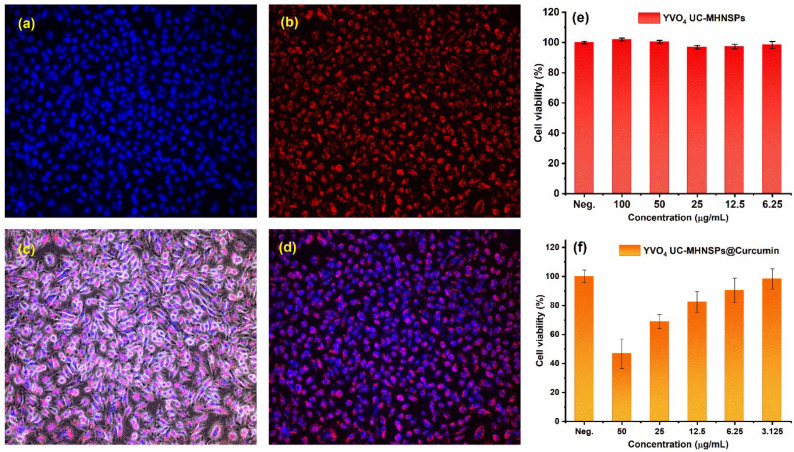
In vitro fluorescence images of YVO_4_:Er^3+^/Yb^3+^ UC-MHNSPs uptake by HeLa cell lines, (**a**) nuclear stain (Hoechst 33342), (**b**) red emission from YVO_4_:Er^3+^/Yb^3+^ UC-MHNSPs, (**c**,**d**) merged fluorescence bright-field and dark-field pictures of HeLa cells under 980 nm excitation. (**e**) Cell viability assessed by CCK-8 assay after incubating with various quantities of YVO_4_:Er^3+^/Yb^3+^ UC-MHNSPs for 24 h and (**f**) antitumor activity of YVO_4_:Er^3+^/Yb^3+^ UC-MHNSPs after conjugation with different quantities of curcumin.

## Data Availability

The data that support the findings of this study are available from the corresponding authors upon reasonable request.

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
