# Peer review of "Evolution of Highly Biocompatible and Thermally Stable YVO4:Er3+/Yb3+ Upconversion Mesoporous Hollow Nanospheriods as Drug Carriers for Therapeutic Applications"

_nanomaterials, 2022, doi:10.3390/nano12152520_

Round 1

Reviewer 1 Report

In the manuscript, the authors reported a highly biocompatible upconversion mesoporous hollow nanospheriods (YVO4:Er3+/Yb3+UC-MHNSPs) as a prospective nano-platform for cancer imaging and therapy. While the topic is generally interesting and ingenious, there are still some problems in manuscript needed to be addressed before further action.

(1) Electron microscope images and nitrogen adsorption curve results indicate that the "hollow mesoporous structure" claimed by the authors seems not have a large specific surface area. If the authors think it's a good drug carrier. The authors should point out the loading amount of curcumin and explain its loading mechanism.

(2) In order to highlight the advantage of YVO4:Er3+/Yb3+UC-MHNSPs as a drug carrier. It is suggested that supply the cytotoxicity differences between YVO4:Er3+/Yb3+UC-MHNSPs loaded curcumin and free curcumin.

Author Response

First, we would like to express our profound thanks to you for your kind encouragement by initial consideration of our manuscript (Manuscript ID# nanomaterials-1805804) and for providing feedback from the reviewers. We welcome the comments to improve the manuscript in a noteworthy way by incorporating suggested revisions in the revised paper.

Reviewer 1.

Query 1: Electron microscope images and nitrogen adsorption curve results indicate that the "hollow mesoporous structure" claimed by the authors seems not have a large specific surface area. If the authors think it's a good drug carrier. The authors should point out the loading amount of curcumin and explain its loading mechanism.

Answer: Thank you for the reviewer comment. Actually, when compared to the mesoporous silica nanoparticles surface area, our synthesized YVO4:Er3+/Yb3+ UC-MHNSPs showed somewhat lesser surface area. Usually, the specific surface area decreases with increasing particle size. However, in the present study, compared to the YC:Er3+/Yb3+ nanospheres of around 50 nm the surface area increased from 25.7 to 87.6 m2/g, which means 3.4-fold increase, for YVO4:Er3+/Yb3+ UC-MHNSPs of around 150 nm and it happened only because of the hollow mesoporous structure of YVO4 particles. Besides, the surface area of the nanoparticles depends on many factors like size, morphology, synthesis method and reaction conditions. Therefore, in my view, the acquired surface area is good enough to use as a drug carrier. Also, the loading mechanism of curcumin was already presented in the subsection “2.4. Drug loading and in vitro releasing protocol” of “2. Materials and Methods section”. Coming to the loading amount of the curcumin, now we have estimated the amount of curcumin loaded on the YVO4:Er3+/Yb3+ UC-MHNSPs and presented it in the revised manuscript. For the reviewer’s convenience, the changes made in the revised manuscript are presented here.

Corrections made in the revised manuscript page No. 8:

             We have also calculated the drug loading efficiency of YVO4:Er3+/Yb3+ UC-MHNSPs and the amount of curcumin loaded on YVO4:Er3+/Yb3+ UC-MHNSPs was estimated to be about 12%. Most of the time, the direct loading of drug does not yield high loading efficiencies and surface functionalization is needed to improve their drug loading efficiency. In the present study, the curcumin was loaded on YVO4:Er3+/Yb3+ UC-MHNSPs particles without any surface functionalization and thus the acquired amount of curcumin loading was somewhat less.      

Query 2: In order to highlight the advantage of YVO4:Er3+/Yb3+UC-MHNSPs as a drug carrier. It is suggested that supply the cytotoxicity differences between YVO4:Er3+/Yb3+ UC-MHNSPs loaded curcumin and free curcumin.

Answer: Thank you for the reviewer's comment and suggestion. In our previous study (DOI: 10.1039/c3dt52692e) we established that the curcumin-loaded mesoporous flowers more effectively facilitated the nuclear localization of curcumin as compared to the free curcumin. Because the mesoporous particles have the advantage of smaller size as compared to the size of curcumin particles in DMSO or any other solvents. Also, it is well-known fact that curcumin exhibits excellent anti-inflammatory, anti-bacterial, and anti-cancer activities however the direct use of curcumin showed inadequate results in preclinical trials because of its rapid metabolism, low bioavailability, poor aqueous solubility, and quick degradation by ultraviolet light, etc. So that, a suitable nanocarrier is needed to formulate the curcumin nanoparticles, which will help to improve its solubility and intracellular effect. However, according to the reviewer's suggestion, we have verified the cytotoxicity of free curcumin and curcumin-loaded YVO4:Er3+/Yb3+ UC-MHNSPs under the same experimental conditions. The free curcumin with 25 mM concentration showed 56% viability whereas 25 mM concentration of the curcumin-loaded YVO4:Er3+/Yb3+ UC-MHNSPs with 12% loading efficiency (i.e. 3 mM concentration of curcumin) showed 68% cell viability, which is due to the high cellular internalization of curcumin-loaded nanoparticles. In order to avoid the reader’s confusion, we did not add in the revised manuscript. The acquired results were presented here for the reviewer’s convenience.

Reviewer 2 Report

This manuscript by Pavitra et al. describes the development of a curcumin-loaded biocompatible upconversion mesoporous hollow nanospheriods that exhibited antitumor activity in vitro. This work is performed well and scientifically sounds correct. Additionally, this manuscript contains valuable information for readers. However, this manuscript needs to be addressed, especially the reason why these nanospheriods did not show any cytotoxicity despite the nanoparticle possessing very high cationic surface properties (over 30 mV of zeta potential).

Author Response

Replies to the Queries received from reviewers on the manuscript “Evolution of highly biocompatible and thermally stable YVO4:Er3+/Yb3+ upconversion mesoporous hollow nanospheriods as drug carriers for therapeutic applications” (Manuscript ID# nanomaterials-1805804)

            First, we would like to express our profound thanks to you for your kind encouragement by initial consideration of our manuscript (Manuscript ID# nanomaterials-1805804) and for providing feedback from the reviewers. We welcome the comments to improve the manuscript in a noteworthy way by incorporating suggested revisions in the revised paper.

Reviewer 2.

Query: This manuscript by Pavitra et al. describes the development of a curcumin-loaded biocompatible upconversion mesoporous hollow nanospheriods that exhibited antitumor activity in vitro. This work is performed well and scientifically sounds correct. Additionally, this manuscript contains valuable information for readers. However, this manuscript needs to be addressed, especially the reason why these nanospheriods did not show any cytotoxicity despite the nanoparticle possessing very high cationic surface properties (over 30 mV of zeta potential).

Answer: Thank you so much for your appreciation and comment. Actually, the surface charge and zeta potential are not the same but they are related. There are few reports signifying that the cationic surface charge is not sufficient to confer toxicity of nanoparticles and also several cationic nanoparticles with high positive zeta potential showed no or weak toxicity. In general, the cytotoxicity of nanoparticles depends on so many factors such as morphology, size, composition, hydrophobicity, surface area in terms of roughness and porosity, binding sites for receptors, adsorption of compounds, cell type, aggregation in culture medium, nitrogen content, and the cell culture and exposure conditions (eg: cell density, particle concentration, medium composition, and temperature). Therefore, in the present study, our synthesized YVO4:Er3+/Yb3+ mesoporous hollow nanospheriods over 30 mV of positive zeta potential did not show any notable cytotoxicity in HeLa cells under our experimental conditions. Please refer to the following publications for a detailed explanation.

  1. Weiss, M., Fan, J., Claudel, M. et al.Density of surface charge is a more predictive factor of the toxicity of cationic carbon nanoparticles than zeta potential. J Nanobiotechnol19, 5 (2021). (https://doi.org/10.1186/s12951-020-00747-7)
  2. Fröhlich E. The role of surface charge in cellular uptake and cytotoxicity of medical nanoparticles. Int J Nanomedicine. 2012; 7; 5577-5591.
    (https://doi.org/10.2147/IJN.S36111)

Based on the above explanations, we believed that the responses could satisfy the editor as well as reviewers for its consideration in the journal “Nanomaterials”.
